# Uncovering the Model and Philosophy of Care of a Psychiatric Inpatient Mother-Baby Unit in a Qualitative Study with Staff

**DOI:** 10.3390/ijerph19159717

**Published:** 2022-08-07

**Authors:** Grace Branjerdporn, Besalat Hussain, Susan Roberts, Debra Creedy

**Affiliations:** 1Mental Health and Specialist Services, Gold Coast Hospital and Health Service, Gold Coast, QLD 4215, Australia; 2Mater Young Adult Health Centre, Mater Hospital, South Brisbane, QLD 4101, Australia; 3School of Nursing & Midwifery, Griffith University, Logan, QLD 4114, Australia

**Keywords:** perinatal mental health, maternal mental health, health service evaluation, service development, model of care

## Abstract

The postnatal period is high-risk time for the first onset and recurrence of maternal mental health disorders. Untreated maternal mental illness can have significant adverse impacts on a woman, her baby, and the wider family unit. For women with mental illnesses that cannot be managed in the community, psychiatric inpatient mother-baby units are the gold standard treatment whereby mothers are co-admitted with their infant for specialist perinatal and infant mental health assessment and treatment. The study explores the model of care and examines the philosophies of care that are used within a psychiatric mother-baby unit. Purposive sampling was used to conduct semi-structured focus group and individual interviews with multidisciplinary staff members at a single mother-baby unit. Themes derived from these interviews were coded into two primary themes and a range of sub-themes. The first primary theme focused on the Model of Care consisting of the following sub-themes: mental health care, physical health care, babies’ care, building mother-baby relationship, fostering relationships with supports, and facilitating community support. The second primary theme centered around the Philosophy of Care comprising of: person-centered care, trauma-informed care, compassion-centered care, recovery-oriented care, attachment-informed care, non-judgmental care, strengths-based care and interdisciplinary care. The model can be used to provide consistency across mother-baby units and to support core capabilities of staff in providing an optimal level of care.

## 1. Introduction

Maternal mental illness is a significant global health issue, incorporating a wide range of disorders from postpartum depression to anxiety, psychosis, bipolar and eating disorders [1,2]. An estimated 10–15% of women in the developed world suffer from mental illness in the first year postpartum, increasing to 20% in developing nations [3,4]. Previous research has found approximately 1–2 per 1000 postpartum women suffer from severe mental illness that requires inpatient treatment [5]. Untreated maternal mental illness has significant effects on the woman, her family, and her baby, particularly in relation to the infant’s future social and emotional development [6]. Perinatal mental illnesses are associated with significant impairments in mother-child bonding, feeding, and infant development [7]. Perinatal psychopathology can lead to abnormal maternal responses and behavior which can impair the infant’s psychological function long-term [8,9,10,11].

The separation of mother and infant postpartum can be a detrimental and traumatic experience for both [9]. Consequently, mother and baby units (MBUs) are recommended as inpatient mental health services in which the mother-baby dyad is admitted together for treatment. Several national guidelines recognize the importance of such specialist perinatal inpatient services for mothers and infants [10,12]. They recommend these services be staffed by specialist perinatal mental health clinicians, providing a full range of therapeutic services [13]. A number of international facilities also champion the approach of mother-infant inpatient treatment, citing significant and sustained improvements across a range of psychiatric disorders [14]. While evaluations of services in developing nations are scarce, studies of low resource MBUs in India found significant clinical and dyadic improvement, in line with UK and Australian findings [15,16]. Previous research comparing MBUs to general psychiatric wards found that patients and staff preferred the MBU, describing superior facilities, treatment, and holistic, perinatally-focused support [17]. Australian patients evaluating their experiences gave overwhelmingly positive feedback, reporting improvements in parenting knowledge and confidence, and a positive change in their own mental health status [18]. Evaluations of MBUs in several countries displayed strong evidence for their effectiveness in treating maternal mental health, while also identifying improved health and relationships for the co-admitted infants [2,19,20]. Prior studies also indicate that admission to an MBU may reduce the number of subsequent mental healthcare admissions, though further research is required to verify this finding [21].

MBUs are considered the optimal model for treating maternal mental health, as co-admission allows for the woman to receive treatment for her mental health, whilst also receiving support in developing parenting skills [10,13,17,22]. Despite the myriad of available literature regarding MBU services, there is currently a lack of a standardized, holistic models of care for use within MBU services, which limits the amount of comparable data that can be collected across MBUs [23]. A further gap in the literature is clarification around the staff’s roles, responsibilities, and experiences within an MBU. The NICE Guidelines for Antenatal and Postnatal Mental Health recommends MBUs be staffed by specialist perinatal mental health staff; however, this has not yet been examined [13]. The current study aims to better understand the roles, requirements, and perspectives of staff by investigating an existing MBU service, the Lavender Mother-Baby Unit.

The Lavender MBU was opened in March 2017 to support new mothers requiring specialized perinatal assessment and treatment. This is a statewide public service, and the first of its kind in Queensland, Australia, established to assess and treat women with severe mental illness in the first year postpartum. The service works holistically by improving maternal mental health whilst fostering positive mother-infant attachment and parenting confidence. The Lavender MBU also includes some allied health professions who do not traditionally have roles in these settings, such as physiotherapy, dietetics, and occupational therapy. Understanding the clinical reasoning and experience of these subspecialties will provide new insights for expanding and creating more roles for these professions in the future. The identification of service provision characteristics, together with staff collaboration and feedback, will enable the development of a standardized model of care tailored to the specific needs of perinatal mental health care patients in inpatient settings.

The study aims to evaluate and characterize the holistic model of care at an MBU to determine its capacity as a template for use within other MBU services. It will also examine the philosophical foundation underpinning the practices of staff within an MBU. The specific objectives of the study are: to better understand Lavender Mother-Baby staff members’ expectations, perceptions and experiences of their role; to explore barriers and facilitators in implementing a specialized perinatal inpatient unit based on the capability model; and to better understand the workforce training and professional development needs of staff.

## 2. Methods

### 2.1. Study Design

A cross-sectional, qualitative research study was conducted using an evolved grounded theory approach [24]. This research study was approved by relevant human research ethics committees (GU Ref No: 2018/964 and LNR/2018/QGC/43989).

### 2.2. Study Setting

The Lavender MBU is an acute mental health inpatient service that provides specialist perinatal assessment and treatment for women with significant mental health difficulties that cannot be managed in the community and their baby(s) [18]. Women eligible for admission require inpatient treatment for a mental health condition that cannot be managed in the community, have an infant under 12 months old, reside in Queensland (Australia), and are not homeless or at risk of homelessness. Women not eligible for admission are those that require detox for a substance or alcohol use disorder, or have a baby with an infectious disease. The service is publicly funded and admits mother-baby dyads, state-wide. The multidisciplinary team consists of medical staff (consultant psychiatrist and psychiatry registrar), nursing staff (mental health nurses and child health nurse), allied health staff (occupational therapist (OT), social worker, psychologist, physiotherapist, dietitian, pharmacist and infant mental health therapist), and managerial/support staff (team leader, service development and research coordinator and administration officer). Individual and group programs are offered to mothers, such as infant massage, sensory modulation, grounding techniques, mother-baby exercise, and healthy lifestyle support. The average length of stay is approximately 21 days, and women with a range of mental health conditions are admitted (e.g., depression, anxiety, postnatal psychosis, eating disorders, bipolar affective disorder, emotionally unstable personality disorder, schizophrenia, mania). A list of usual daily activities provided for patients is outlined in Table 1.

### 2.3. Participants

Participants for this study were multidisciplinary staff (medical, nursing and allied health professionals) working at the Lavender Mother-Baby Unit between March 2017 and 2022. Both clinical and non-clinical staff (e.g., service development and research coordinator, administration officer) were eligible to participate in the study. Fixed-term contract and permanent staff were included in the study, and casual staff were excluded. Twenty-six staff members participated in this study including 10 nursing staff (38.46%), 10 allied health staff (38.46%), 3 medical staff (11.54%), and 3 managerial/support staff (11.54%). Twenty-four staff members (92.31%) were female, and 2 (7.69%) were male. While all allied health, medical and managerial/support staff were recruited, 62.50% of the nursing cohort was sampled. See Table 2 for further details.

### 2.4. Procedure

Staff members were invited to participate through emails and team meetings. Purposive sampling was conducted to ensure all allied health, medical and managerial/support staff were recruited to capture the diversity of perspectives. Nursing staff were interviewed until saturation was reached.

Face-to-face, semi-structured interviews were conducted with staff. Individual and focus group interviews were completed in a private room during a work shift of the staff. At commencement of the interview, informed consent was obtained as participants were briefed about the purpose of the study and the interview procedure. Participants had the opportunity to review the interview questions prior to the interview. Participation was voluntary and non-participation did not affect their working relationship with others. It was reiterated that participants could decline to answer any questions they did not feel comfortable in answering, and cease the interview as required. Interviews were conducted by a trained qualitative researcher independent to clinical care who was an investigator on the research study. Most interviews were conducted in one sitting; however, 3 interviews (11.54%) were conducted over two sittings. In-depth interviews were completed in 30–60 min.

The interview guide was self-constructed and consisted of questions surrounding their role in the MBU, their experience of working in an MBU including rewarding and challenging aspects, unique characteristics of an MBU, and core capabilities such as key knowledge and skills. Interviews were recorded and transcribed verbatim by professional transcription software and a research assistant. Transcripts were deidentified and personal identifying names were replaced with a pseudonym. Transcripts were sent to participants to check for accuracy and with the opportunity to remove any undesired information. Data in approved transcripts were extracted and analyzed using Tie et al. [24] evolved grounded theory approach. In this analytical process, two researchers independently systematically analyzed data in a series of systematic cycles that compared data across interviews to build an integrated ‘theory’ of action [25]. Themes were discussed and agreement was reached, with a third researcher providing consultation when required. This contemporary methodology was selected as it outlines a systematic and analytical process for developing a coherent theory based on the data related to the model of care and philosophy of care in MBUs. Grounded theory identifies and explains how and why people behave in particular ways. The theory-generating process is inductive, moving from the specific to the general to explain phenomena [26,27]. This analytical approach was selected so that the theory generated could apply to other MBUs.

### 2.5. Data Extraction and Analysis

To complete data extraction, concurrent data collection and analysis occurred whereby the research team collected, coded, and analyzed the initial data before further data collection was undertaken. Theoretical sampling occurred from the codes and categories to clarify uncertainties and test interpretations as the study progressed. Initial coding was completed wherein the research team inductively generated as many codes as possible from early data. Intermediate coding was conducted in which codes and categories were reviewed to identify which ones, if any, could refined, or subsumed beneath other categories. Constant comparative analysis facilitated the analytical and iterative process for coding and category development. Memo writing (i.e., reflective interpretive pieces that build an historic audit trail to document thought processes) also occurred to support data analysis. Advanced coding technique was used to facilitate integration of the final grounded theory. In this stage, abstract concepts were reduced into highly conceptual terms that became categories and findings were presented as a set of interrelated concepts. A coherent grounded theory was generated by following “storyline strategy” that facilitated the integration, construction, formulation, and presentation of research findings.

## 3. Results

Based on analysis of the interviews, the grounded theory was developed around two major themes (Lavender Model of Care and Philosophy of Care), with multiple sub-themes and codes (see Appendix A). See Figure 1 for an abstract diagram of the constructs.

### 3.1. Lavender Model of Care

The Lavender Model of Care is a six-dimensional model based on participants’ views of their services, responsibilities, and experiences of working at the mother-baby unit (MBU), including: mental health care, physical health care, looking after baby, mother-baby relationship, fostering relationships, and community support.

#### 3.1.1. Mental Health Care

Providing mental health care to patients was identified to be the primary goal of admission at the MBU: “the key admission goals for mums would be to improve their mental health” (Lauren, Community Registrar).

Mental health care commenced with a comprehensive perinatal mental health assessment which included exploration of a consumer’s trauma history, including experiences during childhood and obstetric history. Clinicians sought to understand: “…what their childhood was like, the experiences they had, and things they’d like to change for their babies” (Quella, Psychologist).

Kate (Psychiatrist) further suggested that assessing for psychosocial risk factors was important to consider during assessment. “It is important for MBU professionals to be able to sort of have the conversation with women about the domestic and family violence, and to have the skills to identify it and ask about it.”

Having a thorough knowledge of perinatal mental health disorders, their underlying causes, associated symptoms, and presentation were identified to be essential attributes for working in an MBU: “…to work at a mother-baby unit, it is necessary to have the knowledge of what are the common mental health conditions that can happen. So having the knowledge base of what are the features of depression, antenatal postnatal depression, all the anxiety disorders. How to identify postpartum psychosis, bipolar disorder, and schizophrenia” (Kate, Psychiatrist).

“If they (healthcare professionals) understand mum and her diagnosis well, then it’s easy to be able to give her the best care that you can because you are aware of what borderline is, how they present; or the presentations of schizophrenia, how they present; or how a postnatal depression looks like” (Amy, Clinical Nurse).

Understanding the impact of a mother’s mental health on her relationship and interactions with her baby was also important, and indicated the level of acuity of her mental illness. “Each person’s intervention is different because the situation is different. For example, we had a lady who had OCD traits, she was very cautious and apprehensive initially because of some compulsion that she felt during baby’s bath time. So, we provided her support and assistance with bathing her baby, just so she doesn’t feel alone and compelled to complete” (Amy, Nurse). Amy furthered “Somebody comes in unwell, that they are feeling that they want to adopt their babies out because they feel that they are not coping because they are unwell. And to see that person even three weeks later to be shocked by their own thoughts, that they even thought that.”

Participants further highlighted that MBU patients may present in times of crisis, so being able to assess for capacity and manage emotional dysregulation was important. “A good understanding of mental health is critical, and the capacity factor in understanding that within mental health acute crisis situations, there is a specific time when our patients will actually have the capacity and the ability to have insight into addressing their needs, so being quite sensitive to when would be appropriate to be addressing specifically identified needs, identified through the psychosocial screening process” (Hanna, Social Worker). “If a woman with a personality disorder experiences emotional dysregulation, we will have one to one intervention with the woman and develop a management plan specific to her” (Tanya, Nurse Team Leader).

Given that a range of individual and group interventions (e.g., sensory modulation, psychological strategies) were offered to patients, having proficiency in executing diverse interventions was deemed to be critical. “We do a lot of sensory work as we get a lot of postnatal depression, anxiety, and personality disorder cases. And sensory modulation is amazing for that population” (Natasha, OT).

“As a psychologist we have to be well versed in a number of interventions such as ACT [Acceptance and Commitment Therapy], CBT [Cognitive Behavioral Therapy], and DBT [Dialectical Behavioral Therapy] and even the relationship like Gottman work is a necessity… We do exposure response prevention for OCD [Obsessive Compulsive Disorder] patients, help with some depression support and anxiety support … to do with cognition diffusion. We do behavioral activation stuff minimizing self-harm or providing safer alternatives. Also do kind of red zone distress and tolerance stuff…Having good understanding of trauma and trauma interventions is important to work at a mother-baby unit. Groups that we run are for managing thoughts, mindfulness, diffusion of thoughts, mindful eating, emotional regulation, distress tolerance” (Quella, Psychologist).

From a nursing perspective, a range of emotional, practical and recreational interventions were conducted to improve mental health. “We do art therapies that they show interest in or something we got equipment for, like we do pampering sessions. Take them for a walk. Sit in the courtyard with them, mainly listening to their concerns and showing them that their feelings are validated” (Amy, Nurse). “In groups, we target advocacy for patients especially those who are under the Act [Mental Health Act 2016], like helping people understand their rights and services that we can provide. We do crisis counselling, grounding strategies or mindfulness strategies. I suppose assessing if people need that sensory support, like if they have done sensory toolkit with OT” (Erika, Nurse).

Staff reported that embedding interventions that other health professionals had recommended, and using both pharmacological and psychosocial interventions were important. “Our main focus becomes like a psychotropic management but there are other non-psychotropic interventions that we use in terms of monitoring or supporting patients. For example, sometimes I tell patients not to use medication for anxiety but to use other non-pharmacological therapy like use a fidget when distressed” (Josh, Pharmacist).

“In order to prepare consumers [patients] to feel comfortable with recommendations, we need to involve a lot of psychoeducation and sometimes do a brief CBT. As well as some mindfulness… and sometimes recommend medications too” (Maya, Community psychiatric registrar).

#### 3.1.2. Physical Health Care

Assessment and treatment of patients’ physical needs was reported to be an important aspect as “physical health is often a big issue with them” (Lauren, Community Nurse). The physiotherapists, Osha and Penny, both affirmed that they assessed post-birth issues related to pelvic floor, bowel and bladder difficulties, and exercise engagement, as well as pain that may affect their mental health.

“We normally go over the pregnancy and how that might have impacted the patient, we focus on their return to exercises, the importance of exercise for their mental and physical health and their pelvic floor…I would normally do a detailed physical assessment on everybody, so check their bladder and bowel, their delivery, their exercise history, their musculoskeletal history” (Osha, Physiotherapist).

“There is a lot of pain, there is a lot of chronic pain, vaginal, incontinence itself, debilitating, there is a lot of emotion that goes with that, there is also a lot of pain that goes hand in hand often with mental health” (Penny, Physiotherapist).

Physiotherapists provided a range of interventions based on the consumer’s interest and stage of recovery to target these physical health issues and promote positive mental health recovery, such as through walking, mother-baby exercise, pilates exercise, psychoeducation and starting a morning routine with exercises.

“We get them back into things they used to enjoy as…they just lock themselves at home and don’t go out, so even though a walk seems very simple, that is very beneficial. Obviously, there is a huge benefit of getting them out and exercising on their mood and so we try to do it every day and try to get them into that pattern of starting the day and waking up, or ending the day. We do a healthy lifestyle group with the dietitian. Regarding return to exercise, the importance of exercise for their mental and physical health, and their pelvic floor.” (Osha, Physiotherapist).

Dietitians focused on supporting the dietary aspects of recovery and providing management of underlying medical comorbidities, such as eating disorders, diabetes, or coeliac disease. “I do dietary assessment which involves looking at nutritional profiles of mother…Depending on the stage of recovery, we look into eating disorders and look into medical complication or food allergies. And we also do nutrition restoration, so if there’s any undernutrition, we do dietary planning” (Finn, Dietitian). Dietitians worked in collaboration with the OT to conduct cooking groups; where the OT will focus on the skills of cooking (e.g., peeling, safety, budgeting) and the dietitian will focus on the nutritional aspects.

Sleep support was also provided to promote favorable mental health. “If needed, we provide protected sleep for the first few days. If they come in and they’re really fatigued and busted, we stand beside them and support them” (Diane, Registered Nurse). Post-caesarean recommendations in line with obstetric direction were also reinforced by the nursing staff, such as “not able to lift baby” or “use a hospital bed” (Lauren, Community Nurse) as mothers may be admitted within the first six weeks post-birth.

#### 3.1.3. Babies’ Care

Participants highlighted that caring for babies was a central element of care as participants “don’t just switch off from babies” (Erika, Nurse). Given the link between poor maternal mental health and difficulties with child health (e.g., sleeping, settling, feeding), nursing staff focused on providing support with these child health issues. “A lot of babies have had reflux and colic, a reason, perhaps, that their babies being so unsettled, and that’s contributed to their mental health decline” (Victoria, Registered Nurse).

“We need an awareness around child health issues because you know often, that can be a precipitating factor in a woman’s current mental health presentation, and so it’s important that we as nurses are able to… offer that advice and guidance. Obviously making sure that it’s evidence based” (Tanya, Nurse Unit Manager).

Interventions related to sleep and settling, such as education around sleep cycles, responsive settling, understanding baby cues, settling strategies and following the baby’s lead, were important aspects of babies’ care. To ensure the safety of the baby when sleeping, co-sleeping is prohibited on the unit. “One agreement that we recently created was the safe sleep agreement regarding mothers and babies not co-sleeping together. We don’t allow that because of sudden infant death syndrome, mothers being medicated and highly sedated, and as mothers may have thoughts of harm towards baby” (Samantha, Research and Service Development Coordinator).

Both allied health and nursing staff highlighted the importance of feeding. “First and foremost, assessment is safety, assessing if baby is getting fed properly and regularly” (Diane, Nurse).

“Ninety percent of child health problems are dietary, often its weaning solids or feeding difficulties” (Jenny, Child Health Nurse). They described the need for educating mothers on breastfeeding, particularly surrounding medication intake, bottle-feeding and introduction of solids. Mothers often seek guidance on when to breastfeed after taking medications. “Mums are guided for things like medication and their interactions with breastfeeding. We have to assess how bubs (babies) are going developmentally in regard to feeding… breastfeeding, formula feeding or introduction to solids” (Finn, Dietitian).

Nursing staff also taught mother-crafting skills to those who lacked confidence. “Some of them have never had anything to do with babies, so the basics, like breastfeeding, sterilizing bottles, interacting with their babies, even baby cares like giving the baby a bath, some of them don’t feel confident with doing that” (Victoria, Nurse). The OT (Natasha) also explained that assessing the safety and quality of activities of daily living, that mother and baby participate in, was an important part of her role. “We do functional task assessment, we go into community with babies, look at, you know, parent’s occupations in the community, public transport, shopping, cooking, or that kind of support”.

As most staff were adult mental health staff, responding to baby physical deterioration and escalating appropriately was also highlighted. “Other core competencies include knowing how to do resuscitation of babies…and to be aware of deterioration and wellbeing in babies, and knowledge of who to contact in such situations” (Kate, Psychiatrist).

Staff identified that assessment of infant development and providing strategies to mothers to promote favorable development, such as through play and baby massage, were crucial interventions. “We look into normal baby development: their physical, emotional, social, language development” (Kate, Psychiatrist).

“My job is to look at developmental assessments. So, I’m concerned about baby’s behavior. I do Ages and Stages, and look into their social and emotional development” (Jenny, Child Health Nurse). Physiotherapists and OTs also described focusing on developmental assessment, play ideas, and tummy time.

Protecting children was reported to be a critical aspect. “We follow Queensland Health guidelines on liaising with Child Safety as we have the sanctuary obligation as mandatory reporting of potential harm to infants” (Tanya, Nurse Unit Manager). Nursing staff played a key role in assessing mother-baby interactions: “With assessments, we realize if mum is being unsafe with the baby, then the Child Safety process is followed” (Diane, Nurse). As the social worker, Hannah highlighted that her role was to educate mothers involved with Child Safety around processes and interventions. “If the infant goes into Child Safety, we arrange conversations with those mothers around the concerns and help the mothers understand the process of Child Safety so they know the focus is of them being reunified and so empowering and advocating for the mum, to basically effect the changes that Child Safety are requesting in order for them to be reunited”.

#### 3.1.4. Building Mother-Baby Relationship

Developing the mother-baby relationship was found to be an important dimension of care. Foundational in their clinical practice was that staff focused on the relationship between the dyad, conceptualizing the mother and baby to be one unit. “There is a lot more at stake in terms of working with two within the one... right at the core, is trying to figure out how they work together. They face difficulties getting to know each other, so we need to work together and help them develop a relationship [with their baby], so we need to nurture that” (Georgia, Dietitian).

Many mothers had difficulty forming loving and warm attachments to their baby due to their mental illness and previous trauma. “Mothers come in with reporting no attachment or very little attachment or highly avoidant of their baby, or express intrusive harm, thoughts and thinking that, well in the words of a patient, that they’re monsters…Their presentation gets in the way of attachment or bonding or normalizing to motherhood, thus having a good understanding and skills to deal with how trauma impacts attachment is important to work at an MBU” (Quella, Psychologist).

“A lot of them [mothers] expressed that they don’t feel anything when they look at their babies, maybe they love their babies, but they don’t feel it…I find it challenging when mums perhaps put themselves first before their babies to the point that baby will be hungry, but they will see their needs first” (Victoria, Registered Nurse).

The multidisciplinary team at the Unit strived to improve the mother-baby relationship: “Sometimes they stay on the unit longer than they might on an acute unit because we’re working on the relationship between Mums and Bubs” (Erika, Clinical Nurse). Quella (Psychologist) stated that, “Mothers are really anxious. At the start, they don’t know how to care for babies, they come in low in confidence. So, we work on improving attachment, developing a loving bond, and closeness between them”.

Nursing staff utilized a range of mother-baby oriented interventions to build the mother-baby relationship, such as: role modelling and coaching of mothercraft skills; play time; providing the voice of the baby to the mothers; encouraging attachment building exercises (e.g., art therapy); and helping mothers understand baby cues during daily activities. “We do role modelling, like settling the bub or feeding or like engaging with babies. We do art therapy where mothers make baby footprints, for example” (Erika, Nurse).

Similarly, allied health professionals employed a range of strategies to help foster maternal-infant attachment, such as “group sessions on attachment and bonding, so touching on Circle of Security concepts” (Hanna, Social Worker), and providing “psychological strategies and tools to build relationship with their babies” (Quella, Psychologist). The Physiotherapist (Penny) encouraged mothers to engage in exercises with their baby as a way to develop a warm attachment.

The OT (Natasha) shared that, as mothers are easily overwhelmed and bothered by things that they can see, hear and touch, she used sensory modulation approaches to help mothers understand their sensory preferences and, in turn, bond with their baby: “The Sensory Profile that I see most in Lavender, is mothers who have sensory sensitivity and particularly to auditory input, visual input, and sometimes touch as well. So, mothers then struggle to connect with babies as the sensory input provided by a baby and their activities is quite overwhelming. Just think about the auditory input of babies crying and the visual input when playing with bright and colorful toys. But by understanding their Sensory Profile, this helps them understand themselves better and then they interact better.”

#### 3.1.5. Fostering Relationships with Supports

Fostering social support from their partners, family, and friends was a pivotal dimension of care, as “we look into the important role that family plays” (Diane, Registered Nurse). “Because mother and infant dyads don’t exist in isolation, they are a part of a much bigger whole. Of course, that whole is those key support people around them, whether that be a partner or family or friend or community” (Hanna, Social Worker).

Family members and other social supports were engaged throughout “admission and care planning, discharge planning, to ensure a holistic and realistic care plan” (Hanna, Social worker). They were also supported with education and resources. “We speak about family members from the very beginning, so from pre-admission we start gathering all the information about family members or supportive people around the woman, and then incorporate them in treatment and strengthen their relationships…We also include family members in, or create, discharge meetings as well; whether based on site or on the phone, we provide them with psychoeducation, resources or like websites” (Tanya, Nurse Unit Manager).

Families and partners were also involved in therapy approaches to support the implementation and awareness of strategies: “We do discuss in meetings the component of why she’s finding things difficult, it is probably because of the way that she processes the sensory information or because of their [partner’s] own sensory processing. It’s just very powerful when they understand this about themselves. It is important for partners to look from the other person’s point of view as to why they are struggling or what are the challenges; this helps them in understanding each other. So, it’s not always about Dad doing everything to make sure the environment is clean and quiet, but it’s also mum figuring out what dad’s needs are as well” (Natasha, OT). Lavender utilized counselling as an important strategy to foster patients’ relationship with their partners. “Mothers are counselled about father’s situation and journey to parenthood; how fathers were impacted by their wives’ mental health condition, how she should communicate about her mental health with dad’s and think about broader family structure” (Kate, Psychiatrist).

“We do modelling with mum and dad of what it looks like to play with their child, engage with their child and build that social emotional connection when they are eating” (Jenny, Child Health Nurse). Recreational family activities are encouraged to foster family relationships: “On weekends where there are mums and dads, they play with their babies and chat with each other. We are hoping to arrange family barbeques” (Victoria, Nurse).

#### 3.1.6. Facilitating Community Support

Facilitating community support was reported to be an important dimension of care to help “set them up for their whole life” (Lauren, Nurse). This included supporting mothers and linkages with community services for transition back to home post-discharge. “We help support the transition to community services or we keep supporting that woman through consultation and liaison, and provide umbrella support for up to three months post-discharge” (Tanya, Nurse Unit Manager).

“I do discharge planning and follow-up in the community to make sure they’re engaged post-discharge as well. I do consultation and liaison with community services state-wide. Can be, for example, writing a letter and advocating for childcare as it is free for 10 weeks post-discharge, or other individual programs for mothers and babies care as they might need physio sometimes” (Isabel, Social Worker).

As a range of needs are identified during admission that cannot be addressed in the limited admission, and require ongoing support, community services are referred to. “Some things can be addressed during the acute admission phase; other things are identified and then recommendations are made for post discharge care, to then facilitate and support the woman or partners or significant others to link in with relevant services down the track” (Hanna, Social worker).

As a Statewide service, telehealth services were utilized as an important tool to connect patients with different services and improve access. “We do seven-day follow-up and also connect them to other services, sometimes we do face-to-face or do telehealth, especially in far areas of the state” (Isabel, Social Worker).

As mothers admitted had complex needs, patients required support from a range of different services (e.g., allied health professionals, non-governmental organizations, mother-infant therapy, domestic violence services, Child Safety, and refugee/migrant women’s services) and were supported to link up with these services.

“We have to link them with a community dietitian if they have an eating disorder. But to find a dietitian, they [the mothers] get triaged, but the dieticians at a public service isn’t always available state-wide, so we have to utilize other avenues such as contacting health precincts in certain catchment areas. We have to make sure that there will be dietitians available in their local community that might be linked with a certain GP” (Finn, Dietitian).

“Sometimes we need private practitioners such as mother-infant therapists, governmental day programs such as Together in Mind and non-government organizations who run peer support programs and Circle of Security. Sometimes other psychological programs, domestic violence prevention services or women’s legal services, a refugee or migrant women’s resource centers, and trauma services need to be arranged” (Isabel, Social Worker).

“We help patients in identifying the community support around them, improving their access to community support, and how to navigate legal systems, in relation to family law, mediation etc., looking at Child Safety and facilitating and liaising with them” (Hana, Social Worker).

Community support services and relevant external stakeholders were also reported to be included in multidisciplinary care reviews and other meetings, and invited to take part in collaborative care planning. “Most of the time we do care planning jointly with other service case managers when we handover patients. We invite them for care reviews and ask them what their role would be post-discharge and we also keep them in the loop post-discharge” (Isabel, Social Worker).

### 3.2. Philosophy of Care

The Philosophy of Care Principles were underpinned by clinical practice and their attitudes towards patients and families. These included: Person-centered care, Trauma-informed care, Compassion-centered care, Recovery-oriented care, Attachment- and family-focused care, Strengths-based care, Non-judgmental care, and Interdisciplinary care.

#### 3.2.1. Person-Centered Care

The treatment and therapeutic programs were guided by patients’ preferences and goals: “…it’s about what client wants…I look at the person and tailoring what it is I have to say to the person’s personality and style rather than being prescriptive” (Jenny, Child Health Nurse).

“Some mothers are really active in their adult life, they want to go to gym, in that case I take them to gym and design gym program for them” (Penny, Physiotherapist).

“We take in consideration consumers’ [patients’] goals and practices and our job is to make them achieve those goals” (Tanya, Nurse Unit Manager).

#### 3.2.2. Trauma-Informed Care

Understanding the mother’s experience with trauma, such as during childhood, recent pregnancy-related difficulties, and other stressors, was highlighted as vital. “We look into the trauma history from the past. It can be very sensitive and that can be very triggering to people, how their childhood was like, if they were happy with their childhood, and whether there were things they’d like to change because it often impacts on how they perceive their goals with parenting…What’s happened in the past with them, how has their pregnancy experience been like for you, and you break that down into trimesters first, second, and 3rd trimester. Did they have any miscarriage, termination, stillbirths?” (Lauren, Nurse).

Recognizing the signs of trauma in the mother’s behaviors and how they interact with their baby was also viewed as significant. “You need to understand the mother’s background and her previous attachment trauma as it helps explain her behavior. For example, a mother may push someone away because they’ve been rejected a lot in the past and they want to protect themselves” (Samantha, Research and Service Development Officer).

Additionally, adopting interventions to address trauma was perceived to be foundational. “Having a good understanding of trauma, and trauma interventions, such as avoiding re-traumatization, is important to work at MBU. And understanding those trauma responses, such as dissociation and avoidance, are important” (Quella, Psychologist).

#### 3.2.3. Compassion-Centered Care

A key philosophy of care was to show compassion and empathy towards every patient, particularly as the patients had been through difficult situations, and to support optimal outcomes in the therapeutic relationship. “With mental health patients we have to be kind, compassionate, and understanding” (Amy, Nurse).

“When you hear their background, you become more compassionate and think, wow, that must be really hard” (Osha, Physiotherapy).

“I try to be very sensitive and always kind because in my 30 years’ experience, I have learned kindness, compassion, and rapport building makes a lot of differences in other people’s lives” (Jenny, Child Health Nurse).

#### 3.2.4. Recovery-Oriented Care

Clinical care was recovery-oriented, as staff held hope for the patients in improving their mental state, maternal-infant attachment, and overall functioning. Staff voiced that they played a role in the recovery process and were heartened by the difference they could observe from start to end of admission. “We become part of that recovery-based journey and you see big differences in patients. They come in depressed and leave functioning and continuing on the road of recovery” (Claire, Nurse).

Staff also noted the positive change in the relationship between the mother and her baby over the course of the admission. “It’s amazing when you see new mums coming in, struggling with having no feelings of love, or no feelings of that real mother-child bond, and then seeing that build in confidence before they leave. They get so organized and recover so much when you look at them at discharge” (Diane, Nurse).

To support recovery, staff focused on the transition back into the community after discharge. “From the physio’s perspective, I focus on getting them back into things they used to enjoy, and things mums want to do post-discharge” (Osha, Physiotherapist).

#### 3.2.5. Attachment-Informed Care

A foundational principle was to “think about mother and baby together, and keeping in mind who is around them to provide support” (Quella, Psychologist), such as the partner, grandmother of the baby, friends and other key support people. “Thinking about the dyad, that mother-infant relationship, and in the context of the family, is imperative to the work we do within a mother-baby unit” (Tania, Nurse Unit Manager). Consideration of the mother, baby and the dyad together was identified as crucial, with staff reporting: “I guess the challenge or difference is that we are working with mother-baby dyad and we have to be mindful of that, we have to consider them both while designing appropriate interventions” (Georgia, Dietitian).

“Within Lavender, I think it is that mother-baby dyad that is unique, so you have to consider both together” (Osha, Physio).

Strengthening the attachment relationship between mother and family was highlighted by Hanna (Social worker) when she explained, “We look into patient’s attachment history and how it is impacting on their current mental health and relationship with their infant and partners, and how it impacts their current health and what interventions could basically improve and strengthen their attachment.”

Staff were also upskilled in working with families through recognized programs: “What we’ve done is, with particularly the nursing staff, they’re all gradually going through, attending the Family Partnership Training through Child Health, and which they’ve said is very beneficial” (Tanya, Nurse Unit Manager).

#### 3.2.6. Non-Judgmental Care

Being non-judgmental was highlighted to be a core attribute for clinicians when interacting with mothers struggling with parenting and who had difficult backgrounds that affected their presentation. “It requires us to be non-judgmental, especially in this unit. It’s the hardest unit because there are babies involved and you do feel protectiveness towards them…They come from all different backgrounds, all different childhoods, different mothers, crafting abilities and you have to be non-judgmental in meeting them, treating them and caring for them” (Claire, Nurse). Another nurse (Victoria) also acknowledged that “definitely, being non-judgmental is a big thing because we see mums from all walks of life”. Clinicians also responded to mothers during individual and group therapy sessions in a non-judgmental manner. “In group settings we discuss a series of topics of interests and encourage them to ask questions and give them answers in a non-judgmental way” (Josh, Pharmacist).

#### 3.2.7. Strengths-Based Care

The Lavender Model of Care was found to be strength-based, where patients’ strengths were harnessed to facilitate recovery. “We look at their strengths and what they do well and try to exploit that towards their recovery rather than focusing on their deficits” (Diane, Nurse).

Staff also felt it was important to support women in their mothering role: the “goal is to empower mums to see their potential as a mother, as a lot of women come in with a lack self-esteem and confidence, so our job is to make them stronger” (Yvette, Nurse).

Women were empowered to manage their mental illness to support their recovery. “How do I live with mental illness and how can I parent with mental illness and so to give them that empowerment, to give them that knowledge and to watch them grow from their mental condition” (Claire, Nurse).

#### 3.2.8. Interdisciplinary Care

The model of care was revealed to be underpinned by an interdisciplinary approach, where professionals from different backgrounds brought their unique knowledge and skills together towards the single goal of a patient’s recovery. Adopting an interdisciplinary way of working was also perceived to be more effective than working in a silo.

“At Lavender, we follow a multidisciplinary team approach, which is really positive. There is lots of sharing, assessments, discussions or working together with the other members of the MDT. Say, for example, combining with the physio for a healthy lifestyle type approach or combining with the OT for functional assessments and community engagement or with social worker, child health nurse, pharmacist, psychiatrist…What we could do together as a team is lot more than what we could do as an individual” (Georgia, Dietitian).

“We basically have a service therapeutic model where all the staff skill set adds on and works much efficiently than what an individual professional can offer” (Hanna, Social Worker).

In particular, the allied health team collaborated closely together. “We have got the whole allied team that works together as a unit across different facets of patient’s recovery” (Claire, Nurse).

A key forum in which staff communicated with each other regarding the care plan, were the community and inpatient case reviews. “Community case reviews happens twice a week, that’s the intake multidisciplinary meeting, attended by consultants, team leaders for the social worker, me, nursing team leader. We discuss case review and intake assessments” (Lauren, Community Nurse).

Regarding inpatient case reviews: “we do case review as well, so you get discussion with the multidisciplinary team. So, after that meeting is finished, I would generally go down and do a mutual help meeting, so that also runs in conjunction with our other Allied Health members” (Penny, Physiotherapist).

## 4. Discussion

This study is the first to qualitatively explore the model of care and philosophy of care undertaken by multidisciplinary clinicians in a psychiatric MBU. The Lavender Care Framework includes six dimensions to the model of care, and eight underpinning philosophical foundations of care. The dimensions of the model of care included mother’s mental health care, physical health care, baby’s care, building the mother-baby relationship, fostering relationships with supports, and facilitating community support. Foundations of the philosophy of care that were identified include: person-centered care, trauma-informed care, compassion-centered care, recovery-oriented care, attachment-informed care, non-judgmental care, strength-based and interdisciplinary care. The Lavender Care Framework provides an overarching practice framework for management of mothers with significant mental illnesses who are co-admitted. The Framework outlines key goals and therapeutic strategies employed from admission to discharge, including assessment, treatment, and discharge planning. The domains also inform the development of a core capability framework for clinicians that highlights knowledge, skills and attitudes considered highly pertinent for multidisciplinary clinicians working in this specialized area.

The dimension of ‘mental health care’ underpins the primary goal of the service, and includes a comprehensive understanding of psychiatric disorders and associated causes, risk factors, assessment and treatment options. Clinicians also considered it imperative to understand the impact these health issues have on the child and support network of the patient. This is in keeping with recommended care for women, who may require a wide range of targeted psychosocial interventions based on their history and needs, as well as support for any partner or family member involved [2]. ‘Physical care’, the second domain, was an important consideration for clinicians. While this aspect is often overlooked in the literature around perinatal mental health, a large population study found that chronic physical conditions increased the risk of perinatal mental illness [28]. These findings highlight the importance of identifying and treating physical conditions in this cohort. Clinicians defined ‘babies’ care’ as feeding, bathing, treating health issues, and generally assessing that the needs of the child are met. Skills such as feeding, bathing and settling babies are taught to mothers, building parental confidence and attachment, which supports and feeds into the fourth domain, ‘building mother-baby relationship’. These domains are both imperative to the treatment of the dyad, given the documented risk to long-term child physical and mental health in cases of perinatal mental health disorders [29]. Targeting mother-infant interactions in the early postnatal period, and promoting bonding and attachment may be the most promising intervention to mitigate child psychopathology safety risks, while also reducing maternal anxiety and depression [7,8]. The final two domains; ‘fostering relationships with supports’ and ‘facilitating community support’ relate to discharge planning, supporting the transition home, and long-term sustainability planning for maternal and child wellbeing. Involvement of families is recommended in the care of perinatal women, and prior research has found that the role of intimate support networks is pivotal to the outcomes of mother and child [30,31].

The model of care elements in the Lavender Care Framework are also consistent with existing patient-centered clinical assessment tools for mothers in the perinatal period (e.g., New Mum Star model, or the Camberwell Assessment of Needs). The New Mum Star, an outcome measure for a nurse-led home-visiting program for first-time mothers in the community, consists of similar domains: health and wellbeing, looking after baby, baby’s development, safety and stability, connecting with babies, relationships, life skills, goals and aspirations, and family and support network [32]. Another model is the Camberwell Assessment of Needs for Mothers (CAN-M) which assesses biopsychosocial needs and support requirement for mothers with mental illnesses during the peripartum period from the perspective of both the patient and the health professional [33]. Congruent with the Lavender Care Framework, the CAN-M covers 26 domains, including psychotic symptoms, psychological distress, self-care, daytime activities, self and child safety, practical and emotional demands of childcare, accommodation, and violence and abuse. While the New Mum Star and the CAN-M are patient-oriented tools, the Lavender Care Framework focuses on staff core capabilities and domains of intervention.

Only one other study has examined the perspectives of staff working in a psychiatric mother-baby unit (*n* = 11) [17]. While Griffith et al.’s study had a different research question of comparing experiences with non-MBU staff and patients, results similarly identified that MBUs conducted parenting interventions, help with infant-care, mother-baby relationship interventions, fostered relationships with the family, and aimed to facilitate community support. Griffith et al. [17] also identified that being compassionate, respectful and non-judgmental were key aspects to building therapeutic relationships.

The philosophical foundations of care highlight the underpinning lens—“ways of being” and attitudes that were considered important by healthcare staff working at MBUs. Analogous to the base of an iceberg underneath the water, the philosophy of care constitutes paradigms that permeate every interaction provided at the MBUs. The staff reported that being patient-centered was important as it meant patients and carers were involved throughout the admission process to support the mother’s recovery and wellbeing [34]. This type of attitude supports collaboration and integration of the patient’s needs as the foremost priority, which is also emphasized in national health service standards [13]. Being trauma-informed is espoused as a key framework for people to acknowledge the impact of trauma, and understand the person’s presentation through this [35]. This is particularly important in an inpatient setting where patients may be presenting with challenging and difficult behaviors, and staff are required to see this in light of their trauma history. Compassion-centered care focuses on empathy and feeling care for the person [36]. Having this mindset supports health professionals to have a humanistic view during their care. Recovery-oriented care [37] emphasizes having hope for patients that they will recover, particularly as patients are highly unwell when initially admitted. Attachment-informed care foregrounds that relationships are integral to human life, and the relationship between the mother and her baby, baby towards the mother, and mother with her family, friends and community are important [38]. Non-judgmental care is a key framework as mothers have severe mental illnesses, and may be initially admitted with negative thoughts and actions towards their baby [39]. While an individual’s natural tendency is to judge the mother negatively, staff consider it important to refrain from making value-judgements. Strengths-based care [40] was recognized as pivotal particularly as mothers may have very complex difficulties but focusing on the positives and the resilience enables mothers to recover. Finally, interdisciplinary care was important as staff had effective relationships with each other in order to complete interventions together and devise comprehensive care plans [41].

The Lavender Care Framework acknowledges the impact that maternal mental illness has on a mother’s relationship with her baby and her practical skills in caring for her baby, as well as treatment approaches to support mothers in these ways. Psychiatric episodes are triggered by childbirth, causing significant morbidity and mortality, including suicide being a leading cause of maternal deaths [2]. This research also highlights the importance of considering and addressing the psychosocial risk factors of perinatal mental illness such as domestic violence, low social support, and personality traits [42,43,44]. For instance, providing support and psychoeducation to partners and including them in the treatment is part of fostering relationships with supports. Such findings inform interventions which mitigate the risk of perinatal mental disorders on psychological and developmental disturbances in children [29,45].

The Lavender Care Framework is in in line with other models. For instance, similar to the Biopsychosocial Model, the biological, psychological, and social characteristics of both the mother and the baby are assessed and treated (e.g., biological: mother’s physical health care, baby’s development; psychological: building mother-baby relationship, mother’s mental health care; social: babies’ care, fostering relationships with supports, facilitating community support). The Framework is also congruent with the World Health Organization International Classification of Functioning, Disability and Health as components of health condition (e.g., mother’s mental health care), body functions and structures (e.g., physical health care), activities and participation (e.g., babies’ care, building mother-baby relationship), environment (e.g., facilitating community support), and personal factors (e.g., mother’s mental health) are recognized as targets during goal setting, care planning and post-discharge support [46]. The Lavender Care Framework is also consistent with the Think Family model which focuses on the bidirectional relationship that exists between the parent and child mental health and its impact on parent-child relationships. [47]. Similarly, the Lavender Care Framework focuses on the mother, baby, and the dyadic relationship between the mother and baby. While the Think Family model identifies that assessing risk and protective factors is valuable, the Lavender Care Framework specifies the importance of the mother’s physical health, family relationship building, and fostering community engagement and support. The Lavender Care Framework also highlights the benefits of adopting a systemic approach [48] as a variety of systems surrounding the mother are considered (e.g., mother, baby, family, community). Domains within the Lavender Care Framework also overlap with the Lavender Recovery Framework which examined care plans from this same MBU [48]. This demonstrates that the consumer’s goals and the clinician’s goals are aligned.

### 4.1. Limitations

While a purposive sampling strategy ensured that a large proportion of nursing staff and all other multidisciplinary clinicians were recruited from the mother-baby unit of interest, only one mother-baby unit’s staff were examined. Despite the Lavender Care Framework being consistent with other models, this may limit the generalizability to other MBUs. While predominantly individual interviews were completed, some nursing staff were interviewed in a focus group with only nurses of varying levels, which may have biased the information obtained. While other units highlight the benefit of peer support workers [17], there is no peer support worker dedicated to the Lavender Mother-Baby Unit, so the perspective of this role was not included. Future research may examine the perspectives of peer support workers.

### 4.2. Clinical and Research Implications

The Lavender Care Framework may be used to support core capabilities of staff working in an MBU. The core capabilities may outline key knowledge, skills and attitudes of multidisciplinary staff in an MBU as this is a specialized area and requires high expertise. Reflecting on these dimensions and philosophical foundations informs selection, onboarding and ongoing training support for staff to provide care that is specific to MBUs. The Lavender Care Framework also ensures that there is comprehensive and high-quality assessment and treatment to aid a mother’s recovery. The Framework has benefits for other MBUs and may be particularly relevant for new MBUs that are being established. Having a consistent model and philosophy of care will support high reliability healthcare and systemic agreement between different units. This is particularly useful for MBUs that share catchments together (e.g., both MBUs have statewide catchment) or will pool data together for benchmarking purposes. This Framework can be embedded in MBU service profiles which are helpful for stakeholders (e.g., patients, the MBU team, external clinicians, hospital executives, funding bodies, and other hospital and health services) to understand what an MBU admission focuses upon. Future studies could also investigate international differences in MBU staff perspectives and examine how we can best evaluate these domains of intervention.

## 5. Conclusions

This study presents a framework that identifies the key dimensions of intervention and philosophical foundations of care that clinicians consider important in the care of women with significant mental illnesses and their baby. This study used qualitative approaches to develop a framework which enables holistic and comprehensive care, is evidence-based, and overcomes shortcomings of existing models. The results highlight six dimensions that inform care planning within an MBU setting and provide a template for setting holistic recovery goals that targets mental health, physical health, baby’s health, mother-baby relationship, and community relationships. Finally, the study reveals philosophical underpinnings of healthcare staff working at MBUs to support care that is respectful, compassionate and oriented around the consumer’s strengths. The Lavender Care Framework is pivotal in supporting the unique needs of mothers with severe mental illness co-admitted with their baby in an acute inpatient setting.

## Figures and Tables

**Figure 1 ijerph-19-09717-f001:**
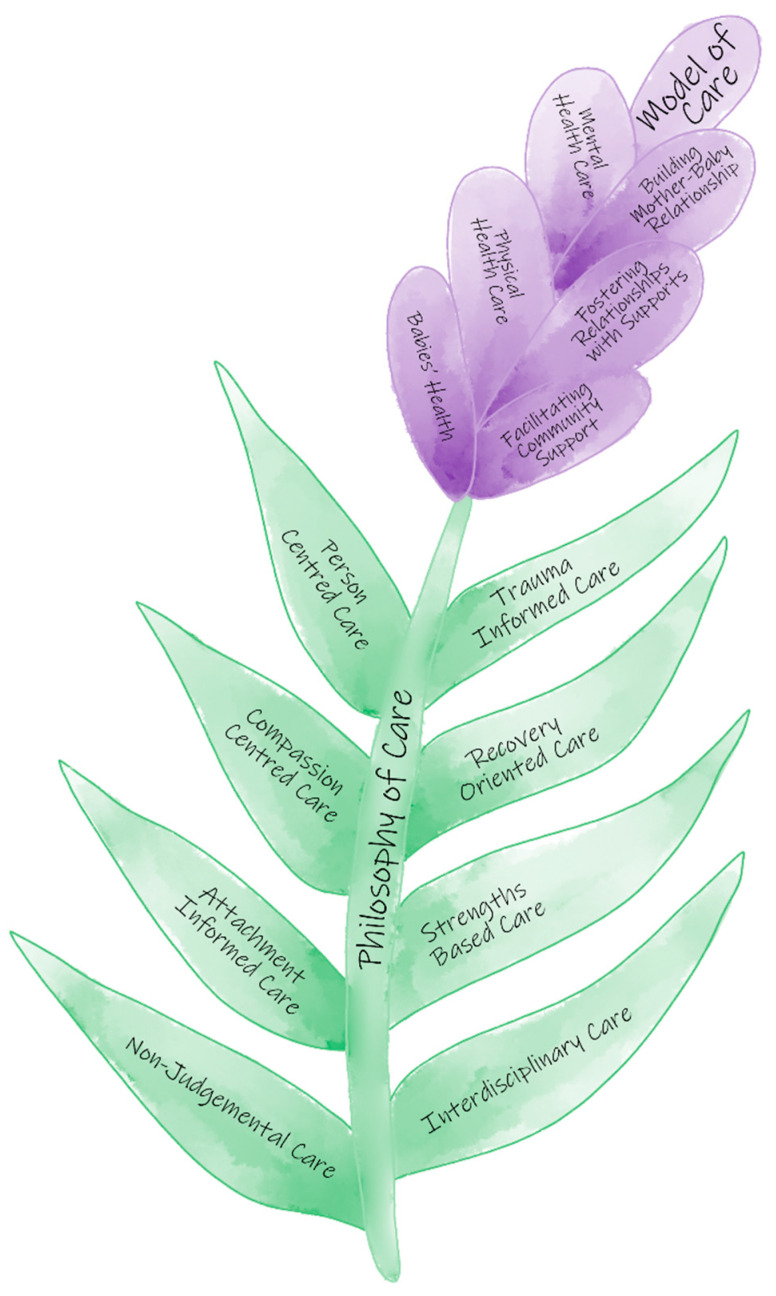
Lavender Model and Philosophy of Care.

**Table 1 ijerph-19-09717-t001:** Example of daily activities provided to patients.

Therapeutic Milieu
Breakfast for mother and baby, with nursing support providing coaching support
Mutual help meeting led by allied health and nursing staff to set the group program for the day and have a round of thanks
Twice daily group program led by allied health and facilitated with nursing staff (e.g., infant massage, sensory modulation, medication education group, Circle of Security, couple communication, positive coping strategies, healthy lifestyles, mother-baby exercise, play groups, and walking group)
Medical reviews, including diagnostic clarification, and prescription of psychotropic medication
Family meeting to support discharge planning
Individual reviews by allied health and child health nurse
Functional/occupational assessment and support by occupational therapist whereby mother and baby go into the community such as go shopping
Meal support by nursing staff for patients with eating disorders
Protected rest-time during the day
Time for mother and baby to be together and enjoy each other
Neurostimulation treatment such as electroconvulsive therapy and repetitive transcranial magnetic stimulation
Evening/weekend activities such as mindfulness, movies, pampering sessions, therapeutic coloring, creative activities, and gardening
Communal dinner promoting peer support
Protected sleep for mother whereby babies sleep in the nursery
Nursing staff providing verbal and practical support and modelling for bath time, feeding, nappy changing and settling

**Table 2 ijerph-19-09717-t002:** Characteristics of clinicians (*n* = 26).

Variable			*n*	%
Discipline				
	Inpatient			
		Mental health nursing staff	7	26.92
		Child health nursing staff	1	3.85
		Psychologist	1	3.85
		Social worker	1	3.85
		Occupational therapist	1	3.85
		Physiotherapist	2	7.69
		Dietitian	2	7.69
		Pharmacist	1	3.85
		Infant mental health therapist	1	3.85
		Psychiatry registrar	1	3.85
	Community ^			
		Nursing staff	1	3.85
		Social worker	1	3.85
		Psychiatry registrar	1	3.85
	Service-wide			
		Nurse unit manager	1	3.85
		Administration officer	1	3.85
		Service development & research officer	1	3.85
		Psychiatrist	1	3.85
Gender				
	Female		24	92.31
	Male		2	7.69

^ Community team completes intake assessments and post-discharge follow-up.

## Data Availability

Data is available in the article. Full, de-identified transcripts may be requested from the corresponding author due to ethical and privacy concerns.

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
