# Peer review of "Uncovering the Model and Philosophy of Care of a Psychiatric Inpatient Mother-Baby Unit in a Qualitative Study with Staff"

_ijerph, 2022, doi:10.3390/ijerph19159717_

Round 1

Reviewer 1 Report

The manuscript addresses clinically relevant issues and is scientifically based. However, the authors could add findings from other sociocultural contexts to highlight the importance of this type of service, as well as its methodological approach to achieving its objectives. 

The authors chose to report the results in a descriptive way, based on examples, but the procedures to reach these findings were superficially described in the method. It would be important to detail the data collection and analysis process.

In the discussion, it would be expected that the authors would resume the objectives of the study and show what its scope is in relation to the existing literature in the area. There was a tendency to highlight the theoretical and practical organizing principles of the Health Unit studied, in a form of circular reasoning, which deserves revision in the argumentation.

It would be important that the authors reflect on the possibilities of transposing this model described to other contexts.

Thus, it is considered that the manuscript may be improved, although our initial opinion is favorable to it. With the suggested additions, probably the manuscript will gain in informative quality, strengthening itself.

Reviewer 2 Report

1.  This study has significance as a study on an important topic for women but not sufficiently addressed until now.

2. Please add the reference for the first sentence of the introduction.

3. In the last part of the introduction, in the sentence introducing the Lavender MBU, the part where only "lavender" is mentioned can be confused by the reader, so please describe it clearly.

4. This study was targeted at the staff of Lavender MBU, please describe the specific inclusion/exclusion creteria. Example: Whether health care provider can participate if they work on MBU, including cases where they have not care patients or have recently worked, etc.

5. Please add the reason for using grounded theory among qualitative research methods.

6. Please move the location of Table 1 to make it easier for readers to see.

7. Please describe who (researcher or assistant) specifically performed the procedure, such as interview, transcribed, etc.

8. In the result, 3.1.1 has the expression consumers. Please clarify if this term refers to patients.

9. Although the introduction stated that marternal mental illness affects women, fer families, and babies, it has not been sufficiently described in the discussion. Please add a section for this.
